# Neural Stem Cell Transplantation for Neurodegenerative Diseases

**DOI:** 10.3390/ijms21093103

**Published:** 2020-04-28

**Authors:** Roberta De Gioia, Fabio Biella, Gaia Citterio, Federica Rizzo, Elena Abati, Monica Nizzardo, Nereo Bresolin, Giacomo Pietro Comi, Stefania Corti

**Affiliations:** 1Foundation IRCCS Ca’ Granda Ospedale Maggiore Policlinico, Neurology Unit, Via Francesco Sforza 35, 20122 Milan, Italy; 2Dino Ferrari Centre, Neuroscience Section, Department of Pathophysiology and Transplantation (DEPT), University of Milan, 20122 Milan, Italy; 3Foundation IRCCS Ca’ Granda Ospedale Maggiore Policlinico, Neuromuscular and Rare Diseases Unit, Via Francesco Sforza 35, 20122 Milan, Italy

**Keywords:** neuronal stem cells, neural subpopulation, neurodegenerative disease, cell therapy

## Abstract

Neurodegenerative diseases are disabling and fatal neurological disorders that currently lack effective treatment. Neural stem cell (NSC) transplantation has been studied as a potential therapeutic approach and appears to exert a beneficial effect against neurodegeneration via different mechanisms, such as the production of neurotrophic factors, decreased neuroinflammation, enhanced neuronal plasticity and cell replacement. Thus, NSC transplantation may represent an effective therapeutic strategy. To exploit NSCs’ potential, some of their essential biological characteristics must be thoroughly investigated, including the specific markers for NSC subpopulations, to allow profiling and selection. Another key feature is their secretome, which is responsible for the regulation of intercellular communication, neuroprotection, and immunomodulation. In addition, NSCs must properly migrate into the central nervous system (CNS) and integrate into host neuronal circuits, enhancing neuroplasticity. Understanding and modulating these aspects can allow us to further exploit the therapeutic potential of NSCs. Recent progress in gene editing and cellular engineering techniques has opened up the possibility of modifying NSCs to express select candidate molecules to further enhance their therapeutic effects. This review summarizes current knowledge regarding these aspects, promoting the development of stem cell therapies that could be applied safely and effectively in clinical settings.

## 1. Introduction

Neurodegenerative diseases, such as Alzheimer’s disease (AD), Parkinson’s disease (PD), Huntington’s disease (HD) and amyotrophic lateral sclerosis (ALS), are highly disabling and ultimately fatal disorders affecting millions of individuals worldwide, with increasing incidence and prevalence [1]. These diseases greatly impact both patients and their caregivers, but no curative therapies are currently available to arrest or reverse their progression. Thus, effective therapeutic approaches are urgently needed.

In neurodegenerative diseases, specific subsets of neurons, such as dopaminergic and cholinergic neurons or motor neurons (MNs), progressively degenerate, resulting in a specific pattern of nervous system dysfunction [1]. Stem cell transplantation seems to be a promising therapeutic choice because of the potential to exert multiple reparative actions within the central nervous system (CNS), including cell replacement and paracrine effects. Different potential stem cell sources are available for therapeutic purposes. Regarding neurodegenerative diseases, the most physiological niche is represented by neuronal stem cells (NSCs), the “seed” cells of the CNS, capable of self-renewing and generating neurons and glia during mammalian CNS development [2]. A small subset of NSCs have been shown to persist after childhood in the subventricular zone (SVZ) of the brain, retaining minimal regenerative ability during adult life [3,4,5]. Due to recent advances in the stem cell field, NSCs can be differentiated directly from pluripotent stem cells, such as human embryonic stem cells (hESCs) and induced pluripotent stem cells (iPSCs), using cell signals and morphogens involved in CNS developmental processes [6].

Remarkably, studies performed into several animal models of different neurodegenerative diseases reported encouraging evidence of a functional benefit after NSC transplantation [7,8,9,10,11,12,13], representing preliminary steps for future clinical translation of this strategy. In this review, we will consider key features of NSCs and related engraftment processes from the perspective of cell therapy approaches for neurodegenerative diseases (Figure 1).

## 2. NSC Properties for Transplanting Purposes

### 2.1. NSC Markers and Subtypes

Different NSC differentiation protocols have been developed to achieve an expandable and well-characterized subpopulation of NSCs from pluripotent stem cells, such as hESCs or iPSCs [14,15,16,17,18,19]. The majority of these multistep protocols (Figure 2) are based on the generation of embryoid bodies (EBs), with subsequent adhesion and expansion in serum-free medium containing growth factors, including fibroblast growth factor (FGF) and/or epidermal growth factor (EGF), nerve growth factor (NGF), brain-derived neurotrophic factor (BDNF), and glial-derived neurotrophic factor (GDNF) [18,19]. The treatment of NSCs with these selected growth factors can stimulate the differentiation of these cells into lineage-committed populations, such as neuronal cells, astrocytes, or oligodendrocytes [18,19]. In other protocols, treatment with small molecules targeting spinal muscular atrophy with respiratory distress (SMARD) signaling promotes neuronal induction, as demonstrated by Chambers’ group [14]. In particular, noggin, a BMP inhibitor, and SB431542, a TGF-β inhibitor, efficiently block the BMP and activin/nodal pathways, promoting the neuralization of human pluripotent stem cells [14].

In-vitro NSC differentiation was tracked at various differential stages with intracellular (Table 1) [20,21,22,23,24,25,26,27,28,29,30,31] and cell-surface (Table 2) markers [15,16,17,28,32,33,34,35]. The cells express paired box 6 (PAX6) and SRY-box transcription factors 1 (SOX1) and 2 (SOX2), transcription factors involved in the self-renewal and maintenance of NSCs in an undifferentiated state [20,23]. Expression of the intermediate filament nestin signals the appearance of neural progenitors and correlates with distinctive properties of progenitor cells, such as multipotency, limited self-renewal and regenerative capacity [25].

Overall, these markers are widely used for the basic characterization of NSCs and neural progenitors; single-cell omics approaches could gain increasing popularity in the future to precisely profile RNA/protein expression in NSCs at a single-cell level.

For transplantation experiments or clinical applications, it may be necessary to purify a specific subset of NSCs or differentiated cells. This selection can be achieved through positive or negative selection for surface markers, taking advantage of fluorescence-activated or magnetic bead cell sorting. Several different analyses of neural clusters of differentiation (CD) antigen expression on NSCs and their derivatives have been performed [15,16,17,28,32,33,34,35] and are summarized in Table 2. For example, negative selection for pluripotent markers, such as SSEA-3, SSEA-4, and the keratin sulfate-related antigens TRA-1–60 and TRA-1–81 [15], can be useful to eliminate undifferentiated cells, reducing the risk of teratoma formation after transplantation [16,17], whereas CD71 could be used to distinguish more mature neuronal derivatives from mixed cultures [32]. Early committed NSCs have been selected for the positive expression of prominin-1 (CD133), SSEA-1 (CD15), or FORSE-1 [32,33]. Meanwhile, differentiated neuronal cells express PSA-NCAM (CD56) and CD24 [16,17], whereas A2B5 [16], CD44 [28], and PDGFR (CD140a) are markers of glial, astrocyte, and oligodendrocyte precursors, respectively [35]. Sundberg et al. [34] evaluated 30 different CD surface markers in TRA-1–81+ hESCs and differentiated neural cells, demonstrating the expression of CD133 and CD326 in undifferentiated pluripotent cells and the expression of CD56 and CD184 in their neural derivatives. The use of these markers has some critical issues; they are not completely specific, and their expression can be influenced by different experimental conditions. Furthermore, cell cycle phases (proliferation or G0), cell source (iPSCs/ESCs), and culture conditions could influence the expression levels of these markers. A possible strategy for increasing specificity is the combination of multiple surface markers using CD antigens (Table 3) [17,28,33].

In 2009, Pruszark et al. [17], combined different CDs, identifying three different populations obtained from hESCs: the CD15(+)/CD29(HI)/CD24(LO) pattern defining NSCs; the CD15(-)/CD29(HI)/CD24(LO) pattern associated with neural crest-like and mesenchymal cell fate; and the CD15(-)/CD29(LO)/CD24(HI) pattern identifying neuroblasts and neurons. FACS selection of the CD15(-)/CD29(LO)/CD24(HI) profile was able to reduce the presence of proliferative cells in human neuronal differentiated ESCs, reducing the risk of in-vivo tumor formation [17]. In a similar broad analysis, Yuan et al. [28] characterized a population of CD184+/CD271–/CD44–/CD24+ neural precursor cells able to differentiate into neurons and glial cells. CD184 is also known as CXCR-4, an alpha-chemokine receptor for stromal-derived factor-1 (SDF-1 or CXCL12), a molecule with potent chemotactic actions [28]. More mature neurons could be selected by matching the CD184–/CD44–/CD15 low/CD24+ markers; instead, the CD184+/CD44+ markers were specific for glial cells [28]. To isolate more stable and defined cell subsets, combining different detection methods could be another possible strategy. More specifically, Turaç et al. [33] developed a screening protocol for identifying surface markers co-expressed on a specific neural cell subpopulation, characterized by validated neural intracellular markers (i.e., nestin, MAP2, doublecortin, and TUJ1). Using this screening approach, researchers identified the CD49f–/CD200+ pattern, a novel surface marker combination to discriminate neural lineages from pluripotent cells. The integrin α6-subunit CD49f is expressed in proliferating cells, while the type-1 membrane glycoprotein CD200, a member of the immunoglobulin superfamily, is an antigen expressed mainly in myeloid lineage cells, neural tissue (principally microglia), vascular endothelium and tumor lines [33]. Beyond the powerful immunoregulatory functions of CD200 and its receptor CD200R [36,37,38], the CD200–CD200R axis has come to be of great interest in neurodegenerative diseases, such as PD [39,40], ALS [41], AD [42] and recently post-stroke inflammation injury [43]. To date, the involvement of this axis in neuroinflammation is still largely unclear, but some evidences suggest that it could act as an inhibitor of proinflammatory microglia factors [44].

Overall, appropriate marker combination also depends on the population needed for the therapy and can vary based on the intended treatment type.

### 2.2. NSC Secretome

Different studies have demonstrated that NSCs are able to increase the survival and regeneration of endogenous neurons by producing neurotrophic factors, such as NGF [11,45,46], neurotrophin-3 (NT3) [11,12,45], BDNF [12,45,47], GDNF [12], ciliary neurotrophic factor (CNTF) [48], vascular endothelial growth factor (VEGF) [11,49], and insulin growth factor (IGF)-1 [50] and 2 [51], which are important modulators of CNS development and function (Table 4) [11,12,45,46,48,49,50,51]. These factors reduce cell death among host endogenous neurons, promoting their axonal/dendritic connection and enhancing transplanted NSC survival and engraftment. The positive effects of NSC transplantation in animal models seem to be mediated by NSCs ability to reduce neuroinflammation [52,53,54].

The “secretome” of NSCs and its correlation with disease improvement in animal models of neurodegenerative diseases has recently attracted the interest of several research groups [11,12,51,55,56,57]. For example, Lee et al. [11] explored the therapeutic effects of human NSCs (hNSCs) obtained from fetal tissues transplanted bilaterally into the lateral ventricles of 13-month-old NSE/APP transgenic mice, an animal model of AD, with the amyloid precursor protein (APP) Swedish mutant allele controlled by the neuron-specific enolase (NSE) promoter. After transplantation, in vitro NSCs produced trophic factors, including BDNF, NGF, and VEGF. Seven weeks after transplantation, NSCs extensively migrated from the injection site and engrafted, differentiating into neurons and glial cells, though the vast majority of NSCs exhibited an immature phenotype. Transplanted mice showed a significant increase in trophic factors that could be correlated to the diminished level of tau phosphorylation detected in these animals. Furthermore, the impaired spatial memory in treated mice was ameliorated 5 weeks after transplantation, without motor alterations or gait impairment, but it did not stop the long-term cognitive impairment. Overall, the complete molecular mechanisms underlying the beneficial action of NSCs in AD mice are unknown, but it is plausible that transplanted cells exert their therapeutic effect by counteracting the toxic signals underlying AD pathogenesis by influencing sites that are relatively distant from the injection point via paracrine effects.

Similar studies have been performed in animal models of PD. Mendes-Pinheiro et al. [12] analyzed the “secretome” of hNSCs (fetal cortical cells) through proteomic analysis, administering NSC-conditioned media to a 6-hydroxydopamine (6-OHDA) rat model of PD and comparing the treated animals to NSC-transplanted animals. The administration of NSC-conditioned media ameliorated pathological hallmarks in terms of motor skills, protecting the substantia nigra and striatal neurons, which was more pronounced than in NSC-transplanted or control animals [12].

To understand the role of the NSC secretome, it is important to identify the factors involved. For this purpose, non-targeted proteomic analyses have been performed using mass spectrometry, identifying previously known neurotrophic factors, such as GDNF and BDNF, and other neuroregulatory molecules, such as PEDF [58], galectin-1 [59], cystatin C [60], clusterin [61], SEM7A [62], and cadherin-2 [63], involved in the migration, differentiation, and neuroprotection pathways. Generally, the majority of proteins secreted by NSCs are functionally related to key cellular and metabolic processes, including antioxidant molecules, TrxR1, Prdx1, and SOD enzymes, which take part in neuron survival [9]. This study was one of the first to analyze the NSC secretome in an unbiased manner [9]. However, additional studies are needed to elucidate the complex molecular signaling regulated by the NSC secretome, a necessary step for clinical translation.

Our group previously demonstrated that the transplantation of NSCs derived from human iPSCs can have a positive therapeutic effect in the context of SMARD1 [10]. After transplantation, NSCs properly engrafted and differentiated in the spinal cord of SMARD1 animals, protecting their endogenous MNs and improving phenotypes. Wild-type iPSC-derived NSCs co-cultured with SMARD1 iPSC-derived MNs ameliorated the pathological phenotype of MNs by producing neurotrophins, including GDNF, BDNF, and NT3, as demonstrated by ELISA on culture medium [10]. Similar results were obtained with ALS MNs and were associated with the same neurotrophins [9,64,65].

All of these studies highlight potentially relevant interactions between transplanted NSCs and the diseased microenvironment of the CNS, which, if appropriately exploited, could modify deleterious toxic or inflammatory states, protecting and promoting tissue regeneration. However, no comprehensive understanding of these events is currently available, particularly regarding immune modulation after human xenografts in animal models. The vast majority of studies rely on an indirect cause–effect correlation to show that transplanted cells have an immunomodulatory role rather than depict specific molecular mechanisms. This knowledge is also limited due to the need to use immunodeficient animals as recipients of xenografts. Another risk that is rarely considered in the field is the potential for opportunistic infections because of immunosuppression, which can influence the experimental setting, even if keeping the animals in ultraclean environments reduces this risk. Therefore, future studies need to precisely assess the interactions among engrafted cells and host tissues to create a more detailed picture of the cellular and molecular events that accompany NSC transplantation. Since the results of hNSCs transplantation in neurological disease models have been promising and have great potential, a thorough analysis of the mechanisms driven by NSCs is necessary to translate this possibility into real clinical therapeutic approaches.

### 2.3. NSC Extracellular Vesicles

Extracellular vesicles (EVs) are nano-sized particles secreted by cells. EVs are composed of cellular membranes with associated membrane proteins, surrounding an aqueous core containing soluble molecules, such as proteins and nucleic acids, including miRNA and mRNA, that can be transferred to local and distant sites [66]. EVs are important effectors of the therapeutic action of NSCs [67]. Upon hNSC transplantation, bidirectional exchange of information through EVs occurs between donor and host cells, promoting the activation of regenerative programs [68].

Despite increasing interest in this field, few studies have been performed to investigate NSC-EV composition and roles. Webb and his group reported beneficial effects of the systemic injection of NSC-derived EVs in in-vivo models of cerebral ischemia, highlighting their anti-inflammatory functions [69,70]. In the first study [60], thromboembolic stroke mice were intravenously injected with labeled hNSC-EVs during the acute phase of stroke and EV biodistribution was evaluated 1 and 24 h after administration using single-photon emission computed tomography. Imaging analysis demonstrated a major distribution of the hNSC-EVs at the level of the injured site in the brain, especially after 24 h. Behavioral tests in hNSC-EV animals, evaluated by the neurological deficit score, demonstrated improved sensorimotor activity with respect to controls. A quantitative flow cytometric study of blood samples from mice 96 h post-stroke was performed to assess the peripheral immune response after EV treatment. The hNSC-EVs enhanced the phenotypic shift of macrophages from a pro-inflammatory M1 pattern to anti-inflammatory M2 features, and treatment promoted the induction of regulatory T cells, reducing pro-inflammatory Th17 cells’ response. Overall, NSC EV treatment in this pre-clinical model resulted in motor and memory amelioration and neuroprotection, augmenting the reparative systemic immune response. In the second study [61], the same group investigated the role of hNSC-EVs in a porcine model of cerebral ischemia using a similar approach. After middle cerebral artery occlusion (MCAO) induction and intravenous administration of hNSC-EVs (2, 14, and 24 h post-stroke), NSC EV actions on tissue level recovery were investigated by MRI 1 and 84 days post-stroke. Brain imaging in stroked pigs showed a significant reduction in edematous lesion area but preserved white matter integrity in NSC EV–treated pigs, confirming the previous results generated by the rodent model [69,70]. Normal behavior and motor activity were also observed. This study shows, for the first time, neuroprotective effects of NSC EVs in a large mammalian model, whose cerebral architecture and white matter composition, similar to humans, is more suitable for translation into human stroke therapies [61]. Similar results have also been observed in a 6-OHDA rodent model of PD after injection of conditioned media from hNSCs, supporting the role of EVs in the secretome [12].

Another study by Rong et al. [71] supports the beneficial effects of EVs treatment in a rat model of spinal cord injury (SCI) by using NSC small EVs (NSC-sEVs). After intravenous injection, motor function evaluation revealed a gradual improvement over the first week for SCI animals treated with NSC-sEVs compared to the untreated SCI group, as demonstrated by BBB scores and gait analysis. MRI confirmed a reduction of lesion region in the NSC-sEVs treated group. Furthermore, NSC-sEVs reduced neuronal apoptosis, as exhibited by significant downregulation of proapoptotic proteins (i.e., Bax and cleaved caspase-3) in both in-vitro (NSC-sEVs pretreated primary neuron culture) and in-vivo (NSC-sEVs treated SCI rats) treated models compared to untreated ones. NSC-sEVs also decreased neuroinflammation and the activated microglia in the injured zone of NSC-sEVs SCI rats, as showed by reduced levels of pro-inflammatory cytokines TNF-α, IL-1β, and IL-6, and reduced levels of CD68-positive cells. These beneficial effects were explained by a strong autophagy induction demonstrated by increased expression levels of two autophagy-related proteins, Beclin-1 and LC3B, in both in-vitro and in-vivo models. The presence of the autophagy inhibitor 3MA in NSC-sEVs pretreated primary neurons significantly reversed the anti-inflammatory and anti-apoptotic effects. These data suggest that NSC-sEV-induced activation of autophagy contributed significantly to apoptosis suppression and neuroinflammatory responses.

Overall, hNSC EVs could represent a cell-free therapeutic tool against neurodegenerative diseases, considering EVs non-invasive administration and their ability to easily cross the BBB without activating an immune response and overcoming some crucial limitations of NSC transplantation (e.g., risk of developing a tumor or malignant transformation and poor durability) [67]. However, isolation and purification procedures to obtain a more homogenous population of EVs in a scalable way are still a challenge, and to date it is not yet clear if hNSC-EVs/conditioned media from different hNSC lines yield the same beneficial effect in treatments of different neurodegenerative diseases. Thus, a depiction of EVs and conditioned media from currently accessible clinical grade hNSC lines is needed to determine the precise secretome profile and analyze the therapeutic potential in animal models of diseases.

## 3. Migration of Transplanted NSCs

The capacity of transplanted cells to migrate from the injection site to a widespread area, or at least within the affected sites, is crucial for a therapeutic effect in the CNS [72]. With respect to developmental stage, in adulthood, the CNS exhibits a restricted regeneration capacity and partially lacks the organized and diffuse radial glia involved into guiding NSCs to the appropriate location [73,74,75]. In many studies, transplanted hNSC grafts create cell clusters close to the site of injection in the adult brain, exhibiting apparently limited migration ability and restricting their therapeutic impact [76]. NSCs express a series of chemokine receptors (e.g., CXCR4) and adhesion molecules that mediate homing to damaged tissue [77]. On the other hand, these chemotactic signals are also produced by NSCs, which can attract each other and cluster together [77]. The limited migration and integration of transplanted hNSCs into the host brain could depend on donor cell differentiation stage, influencing the success of the engraftment. For example, the transplantation of ESC-derived dopaminergic neurons at middle stages of differentiation was suitable for grafting in PD animal models compared to cells derived from earlier or later differentiation stages [78]. Similar results have been shown in a stroke-injured rat model injected with cortically differentiated NSCs derived from iPSCs at different stages of differentiation [79]. Rats engrafted with early and mid-differentiated cells had more graft-derived cells than animals that received more mature cells [79].

To overcome the problem associated with the formation of agglomerates of transplanted neurons, Linaro et al. [80] demonstrated that human cortical pyramidal neurons can be incorporated as single neuronal cells into the neonatal mouse cortex when properly dissociated with ethylene glycol-bis(β-aminoethyl ether)-N,N,N′,N′-tetraacetic acid (EGTA), a chelating substance related to EDTA but with a lower affinity for magnesium, which makes it more selective for calcium ions. In particular, the human neurons create numerous connections with the host neurons and response to sensory stimuli similar to those produced physiologically by host cells [80]. This model could represent a promising experimental system to apply to brain repair and cell therapy strategies [80].

Since limited neuronal migration into host brain tissue seems to depend on preferential chemotactic signaling between the transplanted cells and their differentiated neurons, blocking these signaling molecules allows enhanced migration of human neurons from clusters of transplanted cells [76]. Ladewig et al. [76] demonstrated that factors secreted from neural progenitor cells, such as FGF2 and VEGF, attracted neurons, diminishing their migratory ability. Cells pretreated with FGF2 and VEGF tyrosine kinase receptor inhibitor, with neutralizing FGF2 antibodies or receptor-blocking VEGF antibodies, exhibited higher engraftment and increased spreading of cells 7 days after transplantation into the striatum of adult mice [76].

The differences between developing and adult brains could be another explanation for the relative lack of cell migration in the adult CNS after administration. During CNS development, radial glial cells play a key role in proper migration. When neurons are generated during development, they migrate along the radial glial bundles as a guide to their terminal destination; these bundles disappear almost completely in the adult brain [81]. The CNS environment also presents some differences between the developmental and adult stages [81]. During development, environmental signals aim to enhance neurogenesis and inhibit gliogenesis [81,82,83,84], whereas gliogenic signals are prominent in the adult CNS [82,83,84,85,86,87]. Other elements that could explain the limited plasticity of the adult CNS are the presence of so-called “perineuronal nets” (PNNs), layers of lattice-like brain cells [88]. These PNNs are a broad range of extracellular matrix molecules comprised of chondroitin sulfate proteoglycans (CSPGs) [89]. Interestingly, researchers have discovered that CSPG-degrading proteases can modify PNNs in the presence of some stimuli, such as learning, stress, or CNS injuries [90] Moon et al. [91] demonstrated that partially eliminating PNNs with the chondroitinase ABC, which removes chondroitin sulfate glycoaminoglycans, makes the milieu of the CNS more tolerant to axon regeneration and growth. Therefore, PNN manipulation could improve the migration and integration of hNSCs after transplantation.

Overall, the restricted migratory ability of engrafted cells could be due to a combination of several elements, including the inappropriate maturation status of the injected cells, a lack of guiding signals, and the existence of an inhibitory extracellular matrix that halts cell migration. Therefore, a better understanding of these processes warrants an investigation in order to increase the therapeutic effects of NSCs.

## 4. Enhancing the Therapeutic Potential of NSCs in Neurodegenerative Diseases

### 4.1. Modulation of Axonal Outgrowth

The appropriate integration of neurons derived from grafted NSCs within host circuitries is important to reintegrate functionality in neurodegenerative diseases [92,93]. Transplanted cells must also be able to extend long-range axonal projections [94], which can be difficult in the diseased adult CNS. Consequently, a characteristic that could be modulated to enhance the integration of NSC-derived neuronal cells is their axonal outgrowth towards target neuronal cells. Axonal growth is still present in the adult CNS, although it is reduced compared to the developmental stage. This aspect may be associated with the limited regenerative ability of the adult CNS [95,96]. Thus, various approaches have been explored to overcome this issue and ameliorate axonal extension, including the use of genetically modified hNSCs to overexpress specific factors that promote axonal growth, the treatment of donor cells with compounds that enhance axonal extension before transplantation, and modification of the host environment to make it more permissive to cell migration and axonal growth [93].

One of the main molecules that have been tested is polysialic acid (PSA), a carbohydrate polymer linked to neural cell adhesion molecule (NCAM), which is expressed on neural precursors in both the embryonic and adult CNS [97,98] and plays a key role during CNS development, including neural precursor migration, axonal guidance, and generation of synapses [99,100,101]. Furthermore, a capacity of PSA to lessen cell-to-cell interactions and potentially enhance neuronal plasticity has also been proposed [102]. Battista et al. [102] found that overexpression of PSA in ESC-derived glial precursors remarkably promoted the cellular response to signals guiding migration. These researchers genetically modified donor cells to overexpress polysialyltransferases (PST, the enzymes that synthesize PSA) to increase PSA levels. After transplantation into a rodent PD model, PST-positive dopaminergic neurons derived from murine ESCs had increased graft survival and neuronal process formation (dendrites and axons), augmented axon elongation into the host striatum, and increased synapsin expression, an important marker related to phenotypic improvement [102]. These results indicate that modulation of axon growth can improve transplantation outcome, ameliorating the network integration of transplanted cells and functional recovery.

Axonal regeneration and outgrowth in the CNS seem to be suppressed by several molecules associated with myelin, such as Nogo [103], myelin-associated glycoprotein [104], and oligodendrocyte myelin glycoprotein [105]. Recently, Poplawski et al. [106] pointed out that rat myelin promotes axonal outgrowth from rat NSCs transplanted into sites of damage in adult rats. In particular, they determined that the cell adhesion molecule neuronal growth regulator 1 (Negr1), which is expressed on NSCs, is a key regulator of stem cell-derived axon growth in response to myelin, excluding involvement of Nogo receptor signaling [106]. To overcome the inhibitory action of myelin on axon outgrowth, inhibition of the Cdh1-anaphase-promoting complex (Cdh1-APC), which is present in post-mitotic neurons and pivotal for cell cycle transition, has demonstrated remarkable results [107]. Applying a knockdown strategy in granule neurons of the rodent cerebellar cortex, Cdh1–APC has been reported to selectively inhibit axonal outgrowth, but not dendrite growth. In addition, using an in-vivo RNAi strategy in postnatal rat pups to characterize the action of Cdh1–APC in the regulation of granule neuron axonal outgrowth, Cdh1–APC regulated the growth and patterning of parallel fiber axons of granule neurons in the cerebellum [107].

Several strategies have been explored to improve axonal outgrowth in NSC transplantation. The development of these techniques and a combination of different strategies could be effective in increasing the engraftment rate and graft–host circuitry formation. In addition to the aforementioned mechanisms, several other factors play a relevant role in cell migration and integration, including neurotransmitters [108], neurotrophic factors [47], and inflammatory signals [109,110].

### 4.2. Modulation of the NSC Secretome

Another possible method for enhancing the therapeutic effect of NSCs is modulating the expression of selected growth factors (Figure 3). For example, the genetic modification of NSCs to overexpress IGF-1 [50], a growth factor involved in in vivo neurogenesis and synaptogenesis, has been attempted. Upon transplantation into the hippocampal area of a murine AD model, IGF-1-overexpressing human cortical-derived NSCs exhibited long-term persistence in targeted brain areas, even though no data regarding the impact on disease progression were reported.

NSCs overexpressing neurotrophins (i.e., GDNF, BDNF, NT-3, and NGF) display improved survival and increased proliferative and neuroprotective properties in different neurological disease models [111,112,113,114,115,116,117,118]. In particular, human GDNF-overexpressing NSCs properly migrate towards the disease site and integrate within the CNS after transplantation into the spinal cord of an ALS disease model (SOD1-G93A rats) [118]. Thomsen et al. [117] showed that transplanted cells were able to migrate and differentiate into GDNF-producing astrocytes. Similarly, another group engineered hNSCs ex vivo to express GDNF: these cells survived and secreted GDNF after transplantation in a PD rodent model [119].

In a recent study, Khazei et al. [120] demonstrated that GDNF-expressing human-iPSC-derived NSCs (hiPSC-NSCs) exhibited greater differentiation towards a neuronal phenotype than unmodified control cells after transplantation into a rodent model of cervical spinal cord injury. The expression of GDNF enhanced endogenous tissue improvement and ameliorated the functional integration of engrafted cells, leading to improved neurobehavioral recovery.

BDNF overexpression in hNSCs has also been shown to have similar beneficial effects after transplantation into the contralateral side of the striatum 7 days after MCAO induction in the rat model of stroke [112]. Using MRI, a majority of transplanted NSCs were found to have migrated to the injured area. Neurobehavioral analysis showed significant improvement of these transplanted animals in terms of rotarod tests and modified neurological severity score (mNSS). Engrafted NSCs co-localized with nestin, DCX, and MAP2-positive cells, suggesting that these cells were actively involved in neuronal regeneration and phenotypic improvement. Similar results by Wu et al. [121] confirmed that BDNF overexpression enhances the therapeutic potential of the NSC transplantation strategy in a transgenic mouse model of AD transplanted into the hippocampus of 16-month-old APP transgenic rodents, compared to transplantation of NSCs without BDNF overexpression. Engrafted BDNF-NSC-derived neurons presented the electrophysiological features of mature neurons, appropriately connected with local neuronal circuits, and ameliorated the cognitive deficits of transgenic rodents. BDNF up-regulation enhanced the survival of transplanted cells, improving neuronal differentiation, neurite outgrowth, maturation of electrical features, and synapse formation. In contrast, the reduction of BDNF in BDNF-NSCs diminished the therapeutic impact of stem cell transplantation [121].

Zhang et al. [113] focused on the role of neurotrophin-3 (NT-3) when overexpressed in NSCs via lentiviral vector and administered into the ipsilateral striatum region of rats with stroke. From 2–4 weeks after NSC injection, production of NT-3 protein was significantly more pronounced in transplanted animals than controls, indicating that these genetically engineered cells could survive in ischemic brains and produce NT3 at a high level for at least 2 weeks. This treatment was associated with improved motor functions in treated animals, supporting the potential efficacy of this strategy.

Different timing for growth factor expression having effects on cell engraftment was demonstrated recently by Gantner et al. [122]. Early administration of GDNF promoted the survival and functionality of engrafted human pluripotent-stem-cell-derived dopamine neurons in parkinsonian rats, with plasticity and innervations capability similar to delayed GDNF administration.

### 4.3. Modulation of NSC Differentiation and Functional Properties

The genetic modification of NSCs can also be used to drive the differentiation of transplanted cells, increasing their therapeutic effects. In a rat model of spinal cord injury, Li et al. [123] transplanted Wnt4-overexpressing NSCs; this shifted their differentiation towards a more neural phenotype. Modified cells showed better results in terms of injury repair and functional integration than the transplantation of unmodified ones. Besides this, they also investigated the involvement of β-catenin and MAPK/JNK pathway activation by Wnt4 in neural differentiation.

Another possibility is to genetically modify NSCs to express neurotransmitters that are defective in neurodegenerative diseases. For example, aging and AD features are caused by a reduction in acetylcholine (Ach) neurotransmitter levels. To overcome cholinergic deficiency, Park et al. [46] modified a hNSC line to overexpress the Ach-producing enzyme choline acetyltransferase (ChAT). They transplanted engineered cells into the CNS of an aged rodent model. With respect to unedited cells, ChAT NSCs significantly ameliorated the cognitive function and physical activity of aging animals and induced a significant increase in brain acetylcholine, BDNF, and GDNF. Overall, these findings show that hNSCs overexpressing ChAT produce Ach and restore cholinergic neuronal circuits, improving cognitive function and physical activity in aging mice.

## 5. Requirements and Feature Assessment for Clinical Grade iPSCs

After the desired effect of cell therapeutics is proven in a disease model [124], an appropriate manufacturing process must be established and certified to progress from the research setting to therapy development. However, the characteristics of these cells and methodologies used to assess them lack consensus and are still a matter of debate [125]. Agencies and institutions are joining together to fulfil this need. Among these initiatives, it is worth mentioning the Global Alliance for iPSC Therapies (GAiT), which recently released a set of guidelines.

First, cellular identity should be assessed to avoid cell-line switching and prevent cross-contamination. Short tandem repeat analysis, which should be routinely performed by biobanks for incoming samples, is a suitable tool for this aim and should be performed using a commercially available kit by an accredited laboratory for reproducibility and standardization purposes [125].

Genetic fidelity should be assessed to avoid the transplantation of cells bearing chromosomal rearrangements, which may result in tumorigenesis or unknown side effects. Karyotype analysis is the easiest technique for this purpose, so it should be routinely performed. Complementary single-nucleotide polymorphism (SNP) analysis has also been suggested [125] because it is more sensitive in detecting minor rearrangements and can be reproducibly performed with commercially available SNP arrays.

The absence of a residual reprogramming vector also needs to be assessed. The guidelines suggest testing for at least two different regions of the vector with a quantitative and sequence-specific labeling-based method, such as probe-based quantitative polymerase chain reaction (PCR). The assay should be able to detect less than 1 copy per 100 cells, and it should be optimized with a standard curve comprising previously characterized gDNA [125].

Pluripotency also needs to be characterized, mostly for treatments with cells that are not differentiated but that are expected to undergo terminal differentiation after transplantation. Many techniques are useful for this purpose, including flow cytometry, immunocytochemistry, and quantification of molecular markers of pluripotency. The GAiT guidelines suggest performing flow cytometry to analyze at least two pluripotency markers (e.g., OCT4, Sox2, Nanog, TRA-1-60, TRA-1-81, SSEA-3, and SSEA4) and a combination of intracellular and surface markers is suggested. Regarding pluripotency marker expression, the guidelines point towards the use of a commercially available kit to ensure reproducibility [125].

It is necessary to ensure the sterility of the product before its use; thus, bacteriological and viral testing must be performed to guarantee patient safety [125]. For example, mycoplasma contamination should be ruled out by means of a national pharmacopeia-accepted method. Moreover, the presence of microbial endotoxins and other xenocontaminants needs to be excluded [125].

Measuring other parameters, such as cell viability and doubling time, has also been suggested, because they can provide complementary information.

## 6. Conclusions and Future Perspectives

The absence of resolute therapies for neurodegenerative diseases has a devastating impact on patients and caregivers and places a very high burden on society. NSC transplantation has been explored extensively in the past few decades as a potential therapy for neurodegenerative diseases. In addition to the replacement of lost neurons, stem cells may play a role as neuroprotectors and immunomodulators.

Many preclinical and a few clinical studies have shown beneficial outcomes after NSC transplantation for neurodegenerative diseases. Here, we have outlined and discussed the current concepts of stem cell therapy in neurodegenerative disorders, as well as the most recent developments in this field. Further studies are needed to overcome the actual clinical issues and better understand the translational data for the best exploitation of stem cell therapies in neurodegenerative diseases.

NSCs can be obtained from pluripotent stem cells (iPSCs or ESCs) under conditions that are in line with clinical regulations. The quality of the cell product is relevant and correlates with the standardization of cell harvesting, in-vitro expansion and preparation for transplantation. With the participation of large biotech companies, the use of stem cells may increase due to the accessibility of human iPSC-derived cells obtained by industrialized manufacturing. Establishment of iPSC biobanks would offer the possibility of using well-defined and large-scale produced material in large-scale experiments and/or clinical studies and allowing a high number of concurrent studies.

The structural and functional improvements seen in animal models in the field of neurodegenerative diseases support the therapeutic potential of ESC-NSCs and iPSC-NSCs, encouraging their further research, but further assessment is needed prior to extrapolation to humans.

In this review, we have summarized the key underlying cellular and molecular features important for NSC transplantation and stem cell therapies. To analyze lineage-specific differentiation and isolate specific subpopulations, it is important to apply completely defined in-vitro protocols and eliminate possible artifacts linked with the use of media with scarcely defined composition, including mixtures of a broad range of uncharacterized growth factors. Continuous NSC cultures require quality control. These analyses need to include karyotype and morphological assays, together with control of proliferation and differentiation capacities. To support the stem cell therapy field, optimized and standardized differentiation/selection protocols that allow highly pure populations of NSCs and their differentiated derivatives are urgently needed.

Therefore, it is necessary to decode the fundamental biological aspects of NSCs that control NSC destiny and functional integration after transplantation and to develop safe and effective applications for NSCs in the context of disease. Protein and gene expression profiles derived from omic analyses should be combined with previous knowledge to deliver new biological information about key NSC features. This information and knowledge are essential to define a uniform protein expression profile that is reproducible, predictable, and safe for effective clinical translation of NSCs.

Stem cell therapy can have limitless possibilities if the features, interactions, and functional properties of stem cells are better depicted and controlled. NSCs seem to have potential for the treatment of neurodegenerative diseases, but every stage in the process of clinical application needs serious consideration, from the origin of the cell, through in-vitro cell growth, differentiation, and manipulation, to the final administration to patients. The possible major safety risks posed by stem cell therapy include teratoma formation or abnormal cell growth, immune rejection and inflammation, and undesired cell differentiation phenotypes. These aspects must be carefully controlled. From this perspective, NSC therapy will remain at the top of the research field in the near future, with the aim of providing a response to patients’ urgent needs, directly or indirectly, via regenerative medicine or new molecules.

Overall, we conclude that hNSC-based therapeutic strategies have great potential for the development of treatments for neurodegenerative disorders. However, they are not yet ready for clinical translation; several issues need to be solved, such as the optimal cell source and long-term safety. The need for standardized differentiation/selection protocols and for the generation of cells with reproducible, predictable, and optimal gene and protein expression also have to be properly addressed. The development of strictly defined culture protocols and rigorous quality control may help us achieve this goal. Furthermore, the participation of large companies, which may allow the production of high numbers of cells by standardized industrial manufacturing processes, could further speed translation towards clinical trials. Additional basic studies and phase I/II clinical trials testing hNSCs will also provide more precise and solid knowledge of their mechanisms in the context of disease, supporting their beneficial potential in neurodegenerative disorders. The clinical outcome and long-term safety of stem cell therapy in patients with neurodegenerative diseases are still unclear. The current data from clinical studies testing NSCs, mainly primary cells, will provide information about safety issues, particularly regarding tumor development and the risk of immunorejection. The suitable source of NSCs or more differentiated progenitors for transplantation may be different for different diseases. Evidently, a tailored strategy is needed for each neurodegenerative disease in order to effectively rescue neuronal networks. Knowledge about the capacity of hNSCs to repair and enhance CNS regeneration by neurotrophic and inflammatory modulation, cell replacement, and neuronal plasticity, and whether this capacity can be enhanced by genetic engineering, will allow for more effective and safer stem cell therapy in neurodegenerative diseases.

## Figures and Tables

**Figure 1 ijms-21-03103-f001:**
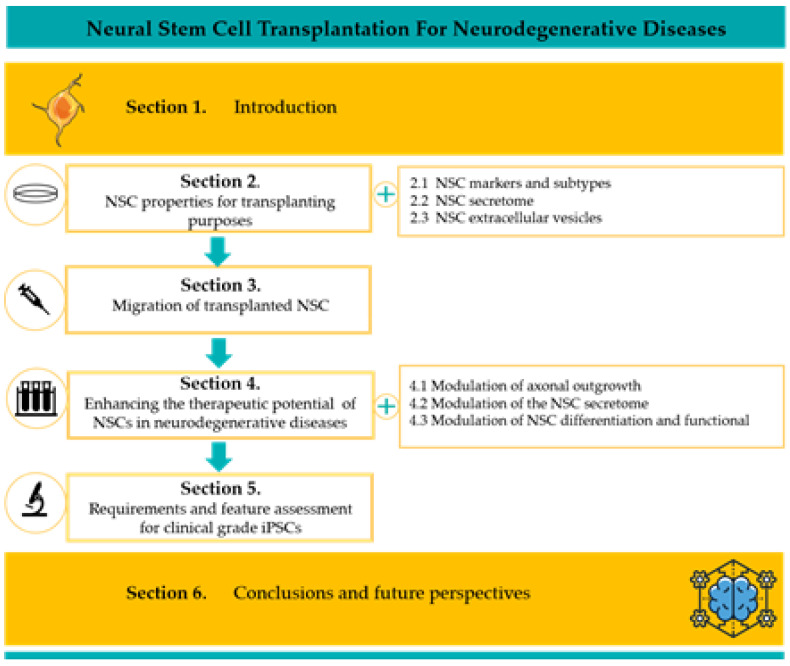
Summary of review’s sections. This review recapitulates some aspects of current knowledge on NSCs (their biological properties in Section 2, their ability to migrate in Section 3 and NSCs-editing strategies to increase their therapeutic outcome in Section 4) for highlighting the strengths and weaknesses of these cells transplantation as a therapeutic strategy.

**Figure 2 ijms-21-03103-f002:**
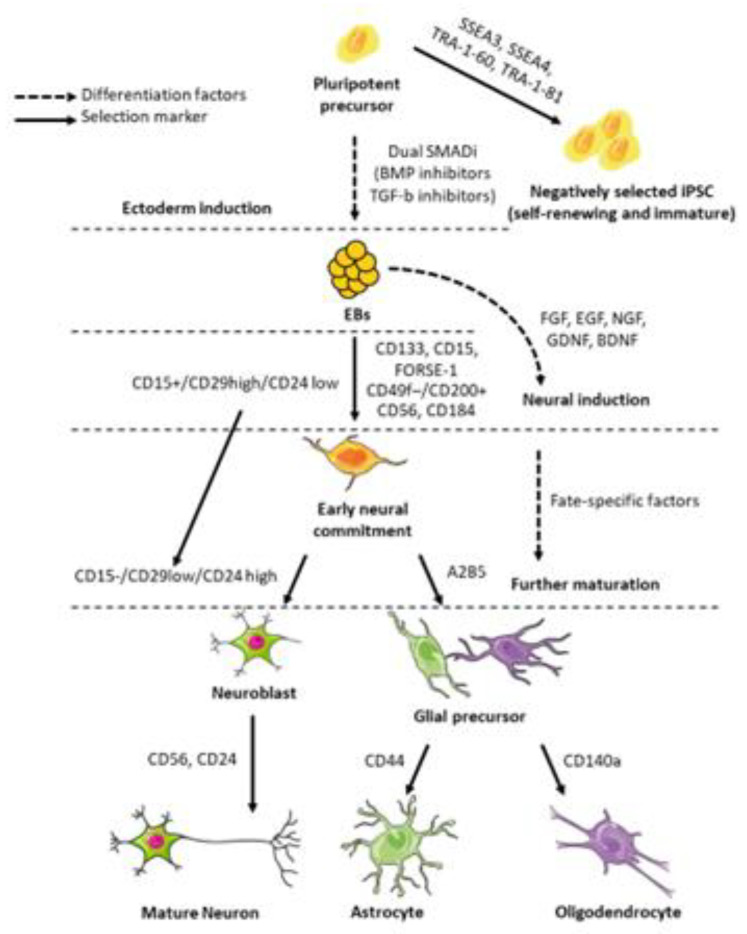
Neural stem cell differentiation pathways and lineage-specific markers. Diagram shows the differentiation stages and defined markers for isolation of NSCs, neurons and glia derived from pluripotent stem cells.

**Figure 3 ijms-21-03103-f003:**
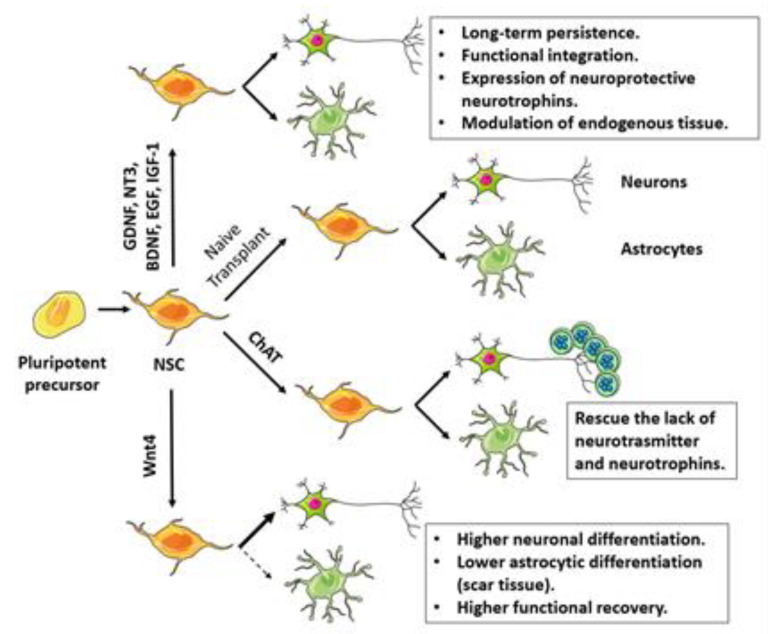
NSCs engineering strategies for transplantation in neurodegenerative disorders and their purposes. GDNF, BDNF, EGF, IGF-1 and NT3 overexpression improved long-term survival and modulated functional recovery after transplantation. Wnt-4 overexpression shifted differentiation toward neural phenotype reducing scar tissue formation and improving functional recovery.

**Table 1 ijms-21-03103-t001:** Intracellular protein expression profile on NSCs and neuronal precursors. Specific sets of markers are useful to track differentiation and discriminate between neural stem cells and committed neural precursors.

	Marker	References
**Neuronal stem cell**	Paired box 6 (PAX6)	[20,21,22]
	SRY-box transcription factor 1 (SOX1)	[21,22,23,24]
	SRY-box transcription factor 2 (SOX2)	[21,22,23,24]
	Nestin (NES)	[21,25]
	Cut-like homeobox 1/2 (CUX1/2)	[22,26]
	Notch homolog 1 (Notch1)	[26,27]
	Hairy and enhancer of split 1/3/5 (HES1/3/5)	[22,23,26,27]
	Cadherin-1/2 (CDH1/2)	[26]
	SRY-box transcription factor 10 (SOX10)	[26]
	Vimentin (VIM)	[22,25]
	Glial fibrillary acidic protein (GFAP)	[22,25]
**Neuronal precursors**	Microtubule-associated protein 2 (MAP2)	[28,29]
	Class III β-tubulin (TuJ1)	[22]
	Doublecortin (DCX)	[22,30]
	ELAV-like protein 3/4 (HuC/D)	[31]
	Neurofilament (NF)	[21,25]

**Table 2 ijms-21-03103-t002:** Progressive expression of cell surface-specific markers. Different sets of genes can discriminate developmental stages as well as lineage specific commitment. After maturation, fate-restricted precursors can be identified by surface markers.

	Marker	References
**Pluripotent cells**	Stage-specific embryonic antigen 3 (SSEA-3)	[15,32]
	Stage-specific embryonic antigen 4 (SSEA-4)	
	T cell receptor α locus 1-60 (TRA-1-60)	
	T cell receptor α locus 1-81 (TRA-1-81)	
**Early Neural Commitment**	Prominin-1 (CD133)	[32,33]
	Lewis X antigen (CD15)	
	Forebrain-surface-embryonic antigen-1 (FORSE-1)	
	Melanoma cell adhesion molecule (CD146)	[16,32]
	P75 neurotrophin receptor (p75)	
	CXCR4, C-X-C chemokine receptor type 4 (CD184)	[28,32]
	Integrin β-1 (CD29)	[32]
	Integrin α-4 (CD49d)	
	Neural cell adhesion molecule L1 (CD171)	
	Epithelial cell adhesion molecule (CD326)	[34]
**Differentiated Neuronal Cells**	Neural cell adhesion molecule (CD56)	[16,17]
	Heat stable antigen (CD24)	
**Glial Precursor Cells**	Neuron cell surface antigen A2B5 (A2B5)	[16]
**Astrocyte Precursor Cells**	Homing cell adhesion molecule (CD44)	[28]
**Oligodendrocyte Precursor Cells**	Platelet-derived growth factor receptor-α (CD140a)	[35]

**Table 3 ijms-21-03103-t003:** Combination of multiple surface markers using cluster of differentiation (CD) antigens. More refined cell type identification can be achieved by a combination of markers, thus excluding partially differentiated cells which may still be able to differentiate aberrantly after transplantation.

Combinatorial Antigens	Cell Phenotype	Reference
CD15(+)/CD29(HI)/CD24(LO)	Neural Stem Cells	[17]
CD15(-)/CD29(HI)/CD24(LO)	Mesenchymal Stem Cells	
CD15(-)/CD29(LO)/CD24(HI)	Neuroblasts and Neurons	
CD184+/CD271–/CD44–/CD24+	Neural Stem Cells	[28]
CD184–/CD44–/CD15 (LO)/CD24+	Mature Neurons	
CD49f–/CD200(HI)	Neural Cells	[33]

**Table 4 ijms-21-03103-t004:** Main growth factors detected in the hNSC-derived secretome. The exploitation of NSCs’ paracrine properties relies on the presence of needed factors in their secretome.

Growth Factor	Reference
Brain-derived neurotrophic factor (BDNF)	[11,12,45,47]
Vascular endothelial growth factor (VEGF)	[11,49]
Glial-cell-line-derived neurotrophic factor (GDNF)	[12]
Nerve growth factor (NGF)	[11,45,46]
Neurotrophin-3 (NT3)	[11,12,45]
Basic fibroblast growth factor (BFGF)	[49]
Epidermal growth factor (EGF)	[49]
Insulin-like growth factor-1 (IGF-1)	[50]
Insulin-like growth factor-2 (IGF-2)	[51]
Ciliary neurotrophic factor (CNTF)	[48]

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
