# Peer review of "Neural Stem Cell Transplantation for Neurodegenerative Diseases"

_ijms, 2020, doi:10.3390/ijms21093103_

Round 1

Reviewer 1 Report

This review is comprehensive, with introduction of extensive knowledge, mostly obtained years and also decades ago but also including recent developments. In view of the importance of the presented information, I expect that interest of the neuroscience community will be considerable, with high numbers of view and reading. My only suggestion is to realise that the review is long, with detailed presentation of numerous similar data of the past. At this point, however, the text can be reduced only marginally. I only recommend an improvement of the english presentation and the general control of the numerous articles presented.

Author Response

Reviewer #1

This review is comprehensive, with introduction of extensive knowledge, mostly obtained years and also decades ago but also including recent developments. In view of the importance of the presented information, I expect that interest of the neuroscience community will be considerable, with high numbers of view and reading.

We thank the Reviewer for positive comments.

My only suggestion is to realize that the review is long, with detailed presentation of numerous similar data of the past. At this point, however, the text can be reduced only marginally.

As suggested by the Reviewer, we reduced the text, summarizing numerous similar data of the past.

 I only recommend an improvement of the english presentation and the general control of the numerous articles presented.

We improved the English presentation with the aid of a professional English editing service and we checked the described articles.

Reviewer 2 Report

Reviewing a review article is difficult and there are no right or wrongs but l have a number of issues with your article.  Overall, l am trying to make suggestions to make it "readable" and interesting for your readers.  At present, it does not meet these requirements as there is a lack of coherence between the sections.  There is a huge amount of literature in this field, and trying to summarize it leads to being superficial in your review.  I think with rewriting, reorganization and better referencing, this could be interesting.  From a readability point of view, the best reviews are those that propose an argument, positive or address a controversy and then review the pros and cons supporting literature to address the issues.  The same could be done in this review where the progress between cell biology and clinical applications can be rigorously considered.  It would be possible to discuss that NSC are ready for clinic (provide supporting studies) or NSC are not ready for clinic and highlight what is needed to make them therapeutic.  The contents in the different sections lack coherence and could be reorganized.  It would be better to have the results from disease models moved into a single coherent section rather than scattered in different sections.

1.  The major problem l have is the lack of definitive descriptions of the work you are trying to review.  There are too many imprecise statements lacking in detail throughout.  Below l will provide some examples.  There is a need to more precisely describe the findings you are discussing.

2.  The article organization, and in particular the paragraph structures, make it hard to read and follow in many places.  For example, in section 3, page 7,line 24-38, why three paragraphs.  These belong in the same paragraph as they are discussing the same reference.  Splitting them into 3 paragraphs is incorrect and very confusing.

3.  It would make for a better paper if the structure could be reorganized to have a figure, or the figure provided, towards the beginning.  The figure should have greater explanation that can be provided in the figure legend.  Starting with a color illustration will attract the reader.  I do not know your labs work, but for review articles, l like to put in a piece of unpublished data to support the story being told.  You could put in a picture of some stained cells to illustrate a key point.  In addition, a flow chart/table describing the different sections and how they are linked would aid the reader.

Specific issues.

1.  Abstract.  Line 13.  "in the majority of cases".  Which cases have an effective treatment.  I would suggest none, but if you can name a disease, please include.  Line 16; delete the first "and".  I would not consider NSCs as a "keyword".  It is an acronym.

2. Introduction.  Line 37, page 1. "other than a few exceptions"  Which ones are you thinking of.  Please provide a reference.

3.  Page 2, line 16.  The section title should be in a different font.

4.  Page 2, line 22. "small molecules".  Please specify molecules so we can begin to under which pathways are being affected.

5. Page 2, line 24.  Explain "neutralization".  I do not understand that term.

6.  Tables.  Please include abbreviations in Table legends.  Makes it easier to follow.  It would be better if you could include a discussion of the papers referenced in these tables in detail.  What are the differences between studies of the expression of these markers.  Do all studies find the same results; are there technical differences between the studies etc.

7.  BDNF and GDNF are neurotrophic factors, not nerve factors (page 3, line 4)

8. Page 3, first sentence.  Same references are needed whenever you state "It has been demonstrated......"

9.  Page 4, line 6.  "In 2007....".  The meaning of this sentence has become lost by its complexity.  Which paper of 2007 are your referencing.  What did they conclude from the profiling.  What type of differentiated neurons were identified.

10. Page 4, line .  "the analysis of single markers does not seem to be sufficient".  Can you make a more definitive statement.  Have anyone concluded which are the best marker combinations for selected cells of therapeutic potential.

11.  page 5, line 8.  What does "widely recognized" mean.  This sentence should be clarified and also defined which of these markers are for which stage of differentiation.

12.  page 5, line 21.  You mention "several papers" but only have one reference.  Words like "attempted to" make the sentence imprecise and can be deleted.  I am sure they did succeeded in finding some secreted proteins.

Similarly, line 34 "likely involve" is imprecise.   Be more specific about what studies have concluded.

13. page 5, line 24.  Define what type of AD transgenic mice.  You are more specific later on.  Need to state the genotypes etc.  Can you discuss ref 11 in more detail - what type of improvements specifically.  

14.  page 5, line 30, 31 are confusing as there are no references.  I am guessing this should be part of the previous paragraph discussing reference.

11.  line 37, should be hydroxydopamine (typo).  Line 39 can you give examples of the "pathological hallmarks ameliorated".

12.  Section 2.2 should be subdivided and include a table of different factors shown to be secreted.  What were the similarities and differences between different studies.  line 42 " Among the factors, they identified
42 some characterized neurotrophic factors, such as GDNF and BDNF".  Were GDNF and BDNF the most abundant, what are the other factors?.

13.  References 45 and 46 deserve more detailed discussion as they are the strongest to provide data on the role of EVs in functional recovery.

14.   Page 7, line 11 onwards.  The bit about EGTA and cell dissociation needs to be revised as there are problems with the sentence.  Further down on line 29 and 32, these should be a single paragraph as the same paper is being referenced.

15. Page 8, line 34. "These researchers ATTEMPTED to genetically modify these cells ....".  Attempted does not seem the correct word.  What did they actually achieve?

16.  Section 4.3.  Can you describe if the genetic modifications in examples presented were better than the unmodified cells.

17.  Overall, the most interesting features could be the effects of secreted proteins, but these are not discussed in details, only in general terms.  I noticed that page 5, lines 3 and 4 you state "POSSIBLY because of the production of neurotrophic factors (GDNF, BDNF, and NT3)"  Imprecise words like Possibly imply that it is not known.  Can you discuss what was known.

18.  Please consider additional figures to highlight your sections.

Round 2

Reviewer 2 Report

It is very difficult to review a manuscript that just shows track changes.  It is better to provide a clean copy as well.  Some journals require it, but if they do not it is my practice to send both versions in the same file.  The clean copy is required to get a thorough (and fair) review.

However, it appears to have been greatly improved. 

1.  The final version needs to be proofed for English again.  With all the scratchings it is hard to see clearly and understand all the sentences that are broken into pieces.  However, some sentences need grammar checking.

2.  The figure legends should have greater detail.  The purposes of the figures in the context of the paper are not apparent.  The formatting of your pdf should be the final version.  The tables and figures have become scrambled in the text.

3.  Better use of subheadings would make the features of this article easier to follow from one section to another.

4.  The section on EVs should be reviewed.  There appear to be little data to support the significance of this but it would be good to consider this area in terms of future potential for therapeutics (or not).

Author Response

It is very difficult to review a manuscript that just shows track changes.  It is better to provide a clean copy as well.  Some journals require it, but if they do not it is my practice to send both versions in the same file.  The clean copy is required to get a thorough (and fair) review.

As suggested by the Reviewer, we send you a clean copy of revised manuscript.

However, it appears to have been greatly improved.

  1. The final version needs to be proofed for English again. With all the scratchings it is hard to see clearly and understand all the sentences that are broken into pieces.  However, some sentences need grammar checking.

We improved the English presentation with the aid of a professional English editing service. As suggested by the reviewer, we rechecked the text and we corrected grammatical errors.

  1. The figure legends should have greater detail. The purposes of the figures in the context of the paper are not apparent.  The formatting of your pdf should be the final version.  The tables and figures have become scrambled in the text.

As suggested by the Reviewer we modified the figure and table legends, adding greater detail.

Figure 1. Summary of review’s sections. This review recapitulates some aspects of current knowledge on NSCs ( their biological properties in section 2, their ability to migrate in section 3 and NSCs-editing strategies to increase their therapeutic outcome in section 4) for highlighting the strengths and weaknesses of these cells transplantation as a therapeutic strategy.

Figure 2. Neural Stem Cell Differentiation Pathways and Lineage-specific Markers. Diagram shows the differentiation stages and defined markers for isolation of NSCs, neurons and glia derived from pluripotent stem cells.

Figure 3. NSCs engineering strategies for transplantation in neurodegenerative disorders and their purposes. GDNF, BDNF, EGF, IGF-1 and NT3 overexpression improved long term survival and modulated functional recovery after transplantation. Wnt-4 overexpression shifted differentiation toward neural phenotype reducing scar tissue formation and improving functional recovery.

Table 1. Intracellular protein expression profile on NSCs and neuronal precursors. Specific sets of markers are useful to track differentiation and discriminate between neural stem cells and committed neural precursors.

Table 2. Progressive expression of cell surface-specific markers. Different sets of genes can discriminate developmental stages as well as lineage specific commitment. After maturation fate-restricted precursors can be identified by surface markers.

Table 3. Combination of multiple surface markers using cluster of differentiation (CD) antigens. More refined cell types identification can be achieved by a combination of markers, thus excluding partially differentiated cells which may still be able to differentiate aberrantly after transplantation.

Table 4. Main growth factors detected in the hNSC-derived secretome. The exploitation of NSCs’ paracrine properties relies on the presence of needed factors in their secretome.

Now, the table and the figures are in the right position in the final version of the text.

  1. Better use of subheadings would make the features of this article easier to follow from one section to another.

As suggested by the Reviewer, we modified subheading in this way:

1.Introduction

  1. NSC properties for transplanting purposes

2.1 NSC markers and subtypes

2.2 NSC secretome

2.3 NSC extracellular vesicles

  1. Migration of transplanted NSCs
  2. Enhancing the therapeutic potential of NSCs in neurodegenerative diseases

4.1 Modulation of axonal outgrowth

4.2 Modulation of the NSC secretome

4.3 Modulation of NSC differentiation and functional properties

  1. Requirements and feature assessment for clinical grade iPSCs
  2. Conclusions and future perspectives

  1. The section on EVs should be reviewed. There appear to be little data to support the significance of this but it would be good to consider this area in terms of future potential for therapeutics (or not).

As suggested by the Reviewer, we added another study to support EVs section:

“Another study by Rong et al [62] supports the beneficial effects of EVs treatment in a rat model of spinal cord injury (SCI) by using NSC small EVs (NSC-sEVs). After intravenous injection, motor function evaluation revealed a gradual improvement over the first week for SCI animals treated with NSC-sEVs compared to the untreated SCI group, as demonstrated by BBB scores and gait analysis. MRI confirmed a reduction of lesion region in the NSC-sEVs treated group. Furthermore, NSC-sEVs reduced neuronal apoptosis as exhibited by significant downregulation of proapoptotic proteins (i.e. Bax and cleaved caspase-3) in both in vitro (NSC-sEVs pretreated primary neuron culture) and in vivo (NSC-sEVs treated SCI rats) treated models compared to untreated ones. NSC-sEVs also decreased neuroinflammation and the activated microglia in the injured zone of NSC-sEVs SCI rats, as showed by reduced levels of pro-inflammatory cytokines TNF-α, IL-1β, and IL-6, and reduced levels of CD68-positive cells. These beneficial effects were explained by a strong autophagy induction demonstrated by increased expression levels of two autophagy-related proteins Beclin-1 and LC3B in both in vitro and in vivo models. The presence of the autophagy inhibitor 3MA in NSC-sEVs pretreated primary neurons reversed significantly the anti-inflammatory and anti-apoptotic effects. These data suggest that NSC-sEV-induced activation of autophagy contributed significantly to apoptosis suppression and neuroinflammatory responses.”

Reference:

Rong, Y.; Liu, W.; Wang, J.; Fan, J.; Luo, Y.; Li, L.; Kong, F.; Chen, J.; Tang, P.; Cai, W. Neural stem cell-derived small extracellular vesicles attenuate apoptosis and neuroinflammation after traumatic spinal cord injury by activating autophagy. Cell Death Dis. 2019, 10, 340, doi:10.1038/s41419-019-1571-8.

Also, we modified the final part of the section, integrating more information about the advantages and concerns of using EVs for potential therapeutic approach.

“Overall, hNSC EVs could represent a cell-free therapeutic tool against neurodegenerative diseases, considering EVs non-invasive administration and their ability to easily cross the BBB without activating an immune response and overcoming some crucial limitations of NSC transplantation (e.g., risk of developing a tumor or malignant transformation and poor durability) [58]. However, isolation and purification procedures to obtain a more homogenous population of EVs in a scalable way are still a challenge, and to date is not yet clear if hNSC-EVs/conditioned media from different hNSC lines yield the same beneficial effect in treatments of different neurodegenerative diseases. Thus, a depiction of EVs and conditioned media from currently accessible clinical grade hNSC lines is needed to determine the precise secretome profile and analyze the therapeutic potential in animal models of diseases.”

Round 3

Reviewer 2 Report

Looks much improved.  The final changes suggested are only minor.

P1. line 23 CNS not SNC

p3, line 5, need greek beta symbol not b

p6 CD200.  As this is a molecule close to my heart, l think describing it as a marker of  neuronal differentiation is not adequate.  Firstly it is a marker whose primary function for immune regulation with CD200R.  Secondly this marker has been shown to be increased in cancer cells with metastatic potential.  It also has been shown to be a marker for other cell types.  A sentence and reference to its immune regulation function would be adequate.  I think CD200 deserves a greater study in relation to neural stem cells and its interaction with immune system.  The reason for high expression in these types of cells is unclear.

Author Response

Looks much improved.  The final changes suggested are only minor.

P1. line 23 CNS not SNC

Thanks for pointing out this mistake. We corrected it in the text.

p3, line 5, need greek beta symbol not b

Thanks for pointing out this mistake. We corrected it in the text.

p6 CD200.  As this is a molecule close to my heart, l think describing it as a marker of  neuronal differentiation is not adequate.  Firstly it is a marker whose primary function for immune regulation with CD200R.  Secondly this marker has been shown to be increased in cancer cells with metastatic potential.  It also has been shown to be a marker for other cell types.  A sentence and reference to its immune regulation function would be adequate.  I think CD200 deserves a greater study in relation to neural stem cells and its interaction with immune system.  The reason for high expression in these types of cells is unclear.

As suggested by the Reviewer, we integrated the sentence with supplement data and additional explanations for CD200 molecule as follows:

“…while the type-1 membrane glycoprotein CD200, a member of the immunoglobulin superfamily, is an antigen expressed mainly in myeloid lineage cells, neural tissue (principally microglia), vascular endothelium and tumor lines [33]. Beyond the powerful immunoregulatory functions of CD200 and its receptor CD200R [36–38], CD200-CD200R axis has developed a great interest in neurodegenerative diseases, such as PD [39,40], ALS [41], AD [42] and recently post-stroke inflammation injury [43]. To date, the involvement of this axis in neuroinflammation is still largely unclear, but some evidences suggest that it could act as an inhibitor of proinflammatory microglia factors [44].”

The following references were added:

  1. Wright, G.J.; Cherwinski, H.; Foster-Cuevas, M.; Brooke, G.; Puklavec, M.J.; Bigler, M.; Song, Y.; Jenmalm, M.; Gorman, D.; McClanahan, T.; et al. Characterization of the CD200 receptor family in mice and humans and their interactions with CD200. J. Immunol. Baltim. Md 1950 2003, 171, 3034–3046, doi:10.4049/jimmunol.171.6.3034.
  2. Gorczynski, R.; Chen, Z.; Kai, Y.; Lee, L.; Wong, S.; Marsden, P.A. CD200 is a ligand for all members of the CD200R family of immunoregulatory molecules. J. Immunol. Baltim. Md 1950 2004, 172, 7744–7749, doi:10.4049/jimmunol.172.12.7744.
  3. Liu, J.-Q.; Hu, A.; Zhu, J.; Yu, J.; Talebian, F.; Bai, X.-F. CD200-CD200R Pathway in the Regulation of Tumor Immune Microenvironment and Immunotherapy. Adv. Exp. Med. Biol. 2020, 1223, 155–165, doi:10.1007/978-3-030-35582-1_8.
  4. Ren, Y.; Ye, M.; Chen, S.; Ding, J. CD200 Inhibits Inflammatory Response by Promoting KATP Channel Opening in Microglia Cells in Parkinson’s Disease. Med. Sci. Monit. Int. Med. J. Exp. Clin. Res. 2016, 22, 1733–1741, doi:10.12659/MSM.898400.
  5. Wang, X.-J.; Ye, M.; Zhang, Y.-H.; Chen, S.-D. CD200-CD200R regulation of microglia activation in the pathogenesis of Parkinson’s disease. J. Neuroimmune Pharmacol. Off. J. Soc. NeuroImmune Pharmacol. 2007, 2, 259–264, doi:10.1007/s11481-007-9075-1.
  6. Chen, Z.; Chen, D.-X.; Kai, Y.; Khatri, I.; Lamptey, B.; Gorczynski, R. Identification of an Expressed Truncated Form of CD200, CD200tr, which is a Physiologic Antagonist of CD200-Induced Suppression. Transplantation 2008, 86, 1116–1124, doi:10.1097/TP.0b013e318186fec2.
  7. Varnum, M.M.; Kiyota, T.; Ingraham, K.L.; Ikezu, S.; Ikezu, T. THE ANTI-INFLAMMATORY GLYCOPROTEIN, CD200, RESTORES NEUROGENESIS AND ENHANCES AMYLOID PHAGOCYTOSIS IN A MOUSE MODEL OF ALZHEIMER’S DISEASE. Neurobiol. Aging 2015, 36, 2995–3007, doi:10.1016/j.neurobiolaging.2015.07.027.
  8. Zhao, X.; Li, J.; Sun, H. CD200-CD200R Interaction: An Important Regulator After Stroke. Front. Neurosci. 2019, 13, doi:10.3389/fnins.2019.00840.
  9. Manich, G.; Recasens, M.; Valente, T.; Almolda, B.; González, B.; Castellano, B. Role of the CD200-CD200R Axis During Homeostasis and Neuroinflammation. Neuroscience 2019, 405, 118–136, doi:10.1016/j.neuroscience.2018.10.030.